# MenatQA: A New Dataset for Testing the Temporal Comprehension and Reasoning Abilities of Large Language Models

**Yifan Wei**[1,2], **Yisong Su**[1,3], **Huanhuan Ma**[2], **Xiaoyan Yu**[1,4], **Fangyu Lei**[1,2],
**Yuanzhe Zhang**[1,2], **Jun Zhao**[1,2], **Kang Liu**[1,2,5]

[1]The Laboratory of Cognition and Decision Intelligence for Complex Systems, CASIA
[2]University of Chinese Academy of Sciences, [3]Fuzhou University
[4]Beijing Institute of Technology, [5]Shanghai Artificial Intelligence Laboratory
{weiyifan2021,mahuanhuan2021,leifangyu2022}@ia.ac.cn, 221020042@fzu.edu.cn
{xiaoyan.yu,yzzhang,jzhao,kliu}@nlpr.ia.ac.cn

## Abstract

Large language models (LLMs) have shown nearly saturated performance on many natural language processing (NLP) tasks. As a result, it is natural for people to believe that LLMs have also mastered abilities such as time understanding and reasoning. However, research on the temporal sensitivity of LLMs has been insufficiently emphasized. To fill this gap, this paper constructs **M**ultiple **Sen**sitive **F**actors **T**ime QA (**MenatQA**), which encompasses three temporal factors (scope factor, order factor, counterfactual factor) with total 2,853 samples for evaluating the time comprehension and reasoning abilities of LLMs. This paper tests current mainstream LLMs with different parameter sizes, ranging from billions to hundreds of billions. The results show most LLMs fall behind smaller temporal reasoning models with different degree on these factors. In specific, LLMs show a significant vulnerability to temporal biases and depend heavily on the temporal information provided in questions. Furthermore, this paper undertakes a preliminary investigation into potential improvement strategies by devising specific prompts and leveraging external tools. These approaches serve as valuable baselines or references for future research endeavors.

**Context:** [1] *Twitter was created by Jack Dorsey, Noah Glass, Biz Stone, and Evan Williams in March 2006 and launched in July of that year.* [2] *On October 16, 2008, Evan Williams became the CEO, and Dorsey became the chairman of the company.* [3] *Jack Dorsey rejoined Twitter in* **March 2011** *as Executive Chief of Product Development.* [4] *In June 2020, Twitter announced that Patrick Pichette would succeed Omid Kordestani as chairman.* [5] *In* **November 2021**, *Jack Dorsey stepped down as CEO and was replaced by Parag Agrawal, the chief technology officer.* [6] *On October 27, 2022, business magnate Elon Musk acquired Twitter for US$44 billion, gaining control of the platform.* [7] *On May 12, 2023, Musk announced that he will resign as CEO of Twitter in approximately six weeks.*

---

*Multiple Sensitive Factors Time QA*
**Scope Factor:**
*Who was the CEO of Twitter from* *May 2013* *to* *2020* *?*
**Order Factor:**
*Shuffle the order of the sentences* *[1-7]* *in the context .*
**Counterfactual Factor:**
*Who was the CEO of Twitter from March 2011 to* *July 2022* *, if Jack Dorsey stepped down as CEO in* *November 2022* *?*

---

**Answer:** *Jack Dorsey*

Figure 1: Yellow front indicates the time specifiers of events in the context. Scope Factor refers to the time specifiers would be different between the question and the given context. Order Factor is where the complete events in the context are shuffled in chronological order. Counterfactual Factor is a question with hypothetical propositions.

## 1 Introduction

Recent Large Language Models (LLMs; Zeng et al. 2022; Touvron et al. 2023; Zhang et al. 2022) such as GPT-4 (OpenAI, 2023) pretrained on a vast amount of text corpus have achieved nearly saturated performance on most Natural Language Processing (NLP) tasks. Meanwhile, plenty of works have evaluated the reasoning abilities of LLMs on several tasks, such as numerical reasoning (Chen et al., 2022), logical reasoning (Saparov and He, 2022), counterfactual reasoning (Li et al., 2023), and multi-hop reasoning (Lei et al., 2023; Ma et al., 2023). However, the temporal reasoning ability of LLMs, which refers to the capacity of a model to

capture the temporal scope and interval of events in a given context, is yet seldomly explored. This ability is particularly important and necessary in many downstream tasks, such as Question Answering (QA), due to the inconsistency of answers in real events across time ranges. For example, as shown in Figure 1, given the context *"From March 2011 to November 2021, Jack Dorsey rejoined Twitter as CEO, and In November 2021 Jack Dorsey stepped down as CEO and was replaced by Chief Technology Officer Parag Agrawal"*, the answer to the question *"Who was the CEO of Twitter from year A to year B?"* could be either *"Jack Dorsey"* or *"Parag Agrawal"*, depending on the time pe-

riod ([**year A** , **year B**]) in the question.

To verify the temporal reasoning ability of models, a few datasets have been proposed. SituatedQA (Zhang and Choi, 2021) focused on how answers vary according to different extra-linguistic contexts, such as, when questions are asked. RealTime QA (Kasai et al., 2022) was proposed to answer questions where real-time news was served as the contexts. Recently, TimeQA (Chen et al., 2021) was proposed for time-sensitive question answering , particularly for the temporal **scope factor**. In TimeQA, the time specifiers would be inconsistent between the question and the given context. As shown in Figure 1, a time specifier in *"Who was the CEO of Twitter from May 2013 to 2020?"* is **2020**, but the time specifier of the correct answer in the context is **November 2021**. As a result, the system needs more powerful abilities of time understanding and temporal reasoning.

Nevertheless, there are more temporal reasoning abilities (factors) that need to be verified but are usually neglected, besides the identified temporal **scope factor** in TimeQA. The first is **order factor**. For example, *"[On October 16, 2008], Evan Williams became the CEO, and Dorsey became the chairman of the company. Jack Dorsey rejoined Twitter in [March 2011] as Executive Chief of Product Development"*. In this example, the chronological sequence of events is laid out by the given time specifiers, illuminating the progression of roles within the company. Consequently, recognizing the chronological order of events is a fundamental ability and typically assessed in evaluations concerning time understanding and reasoning.

The second is **counterfactual factor**. Questions with temporal assumptions greatly escalate the difficulty of temporal reasoning. Answering such questions may require additional information or counterfactual thinking of models. For example, *"Who was the CEO of Twitter from March 2011 to July 2022, if Jack Dorsey stepped down as CEO in November 2022?"*. LLMs should be able to understand that *"Jack Dorsey was still the CEO of Twitter from March 2011 to November 2022"*. Obviously, answering such types of questions is another form to test temporal reasoning ability.

To facilitate the development of research around the aforementioned problems, this paper proposes a new dataset, **M**ultiple **Sen**sitive **F**actors **T**ime QA (**MenatQA**), which encompasses the above three temporal sensitivity factors and is used to evaluate the temporal reasoning ability of the LLMs. In detail, the MenatQA dataset contains 2,853 samples, which are partitioned into 1,448 samples for the **scope** type, 857 samples for the **order** type, and 548 samples for the **counterfactual** type, respectively.

Based on the proposed MenatQA, serveral mainstream models are evaluated, including the SOTA temporal reasoning models (BigBird (Zaheer et al., 2020) and FiD (Izacard and Grave, 2020)), and current typical large language models such as LLAMA (Touvron et al., 2023), OPT (Zhang et al., 2022) and GPT-3.5 (gpt-3.5-turbo; Brown et al. 2020). The experimental results demonstrate the majority of LLMs perform poorly on our MenatQA dataset. It indicates a potential deficiency in LLMs' comprehension of temporal concepts. Moreover, to enhance the temporal reasoning ability of LLMs, especially for aforementioned scope factor, order factor, and counterfactual factor, this paper proposes some preliminary investigations, such as designing specific prompts and tool learning. These approaches will serve as baselines in MenatQA and can be used as a benchmark for future research.

Our main contributions are summarized as follows:

- We present a new dataset named Multiple Sensitive Factors Time QA (MenatQA). This is the first dataset containing multiple time-sensitive factors that can be used as an evaluation benchmark for assessing the time understanding and reasoning abilities of LLMs.

- We evaluate the performance of current LLMs on three temporal factors, revealing their high susceptibility to temporal biases and their reliance on specific temporal information given in questions for reasoning about time.

- We provide preliminary investigations to optimize temporal reasoning ability of LLMs, which can be used as baseline to inspire the future research.

## 2 The MenatQA Dataset

### 2.1 Dataset Construction

**Data collection.** We construct MenatQA based on TimeQA (Chen et al., 2021) dataset. Only time questions that are accompanied with a golden context and a detailed time scope of event are collected. To extract the relevant time scope, correct

| Type | Questions | Answerable | Unanswerable | Doc-Token | #Question-Token |
|---|---|---|---|---|---|
| Scope | 1448 | 1316 | 132 | 227 | 16 |
| Order | 857 | 669 | 188 | 264 | 24 |
| Counterfactual | 548 | 462 | 86 | 227 | 30 |
| Total | 2853 | 2447 | 406 | - | - |

Table 1: The dataset provides statistics for different types of factors. #Doc-Token and #Question-Token represent the average number of tokens within the document and question, respectively. This paper counts the number of tokens using GPT-2 tokenizer, which is the same tokenizer as ChatGPT.

answers, annotated paragraphs, and golden context from documents, we develop a script that utilizes JSON syntax for accurate identification.

**Data annotation.** We represent the time factor as three types: *scope* factor, *order* factor, *counterfactual* factor. The detailed information about the annotation can be found in the Appendix A.2.

- The definition of the scope factor refers to the time scopes that are relevant to the question (e.g., *"From 2011 to 2021"*). Specially, the scope type includes two types of questions: extraction and reasoning. The extraction questions originate from TimeQA[1], and the reasoning questions can be obtained by adding more fine-grained information such as months (e.g., *"From March 2011 to November 2021"*), narrowing the time range (e.g., *"From 2013 to 2020"*), or expanding the time range (e.g., *"From 2008 to 2021"*). In detail, the reasoning type questions are addressed using OpenAI's text-davinci-003 API, employing few-shot learning to alter the temporal intervals mentioned in the questions. Subsequently, we provide both the original and altered questions to the three annotators, requesting them to provide answers to the altered questions based on the contextual information.

- The *order* factor pertains to the chronological sequence of events in the context. Typically, the descriptive information on each Wikipedia page is written in chronological order, as shown in the context in Figure 1. We asked three annotators to read the context, identify different events based on time, and then shuffle the order of these events in the context.

- The *counterfactual* factor refers to hypothetical propositions about time, where the assump-

---

[1]We adopt the Easy-Mode version of TimeQA, which only involves extraction type questions.

tion goes beyond the context and requires imagination to connect the context and the hypothetical question (Li et al., 2023; Tang et al., 2023). Counterfactual questions consist of a question (*"Who was the CEO of Twitter from March 2011 to July 2022?"*), alongside a premise that contradicts the given context (*"If Jack Dorsey stepped down as CEO in November 2022"*). Based on this premise, an imaginary consequence of the counterfactual question yields *"Jack Dorsey"*, as shown in Figure 1. Inspired by previous work on constructing counterfactual samples (Li et al., 2022), we ask the annotators to imagine a temporal hypothesis that contradicts the context (e.g., changes in years). Then constructing a *"if"* question based on the hypothesis, while providing the correct answer. To ensure the diversity of phrasing, annotators are free to generate various phrasing of the assumption, and there is no restriction on the position of the assumption.

## 2.2 Dataset Statistics

**Key statistics.** The MenatQA dataset contains 2853 time-sensitive factor samples, which are partitioned into the *scope* type, *order* type and *counterfactual* type corresponding to 1448, 857 and 548 samples. The main statistical data for factors are shown in Table 1. To address the issue of potential illusory outputs in LLMs, introducing unanswerable questions serves as an effective means to assess their understanding of temporal knowledge. In MenatQA, we find that there are only 85.7% of the questions are answerable questions, while 14.2% are unanswerable questions.

Specially, the scope type includes two types of questions: reasoning and extraction, with 450 and 998 samples, respectively. The extraction type refers to questions where the corresponding time specifier can be directly found in the context, while the reasoning type refers to questions where there

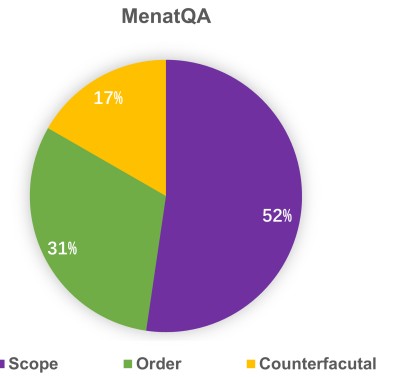

Figure 2: Statistics on the types of time-sensitive factors in the MenatQA dataset.

is a discrepancy between the time in the context and the question. The proportion of time factor types is shown in Figure 2. These statistics indicate that MenatQA exhibits rich diversity in question distribution. The average length of questions in MenatQA is 20.71 words, while the context consists on average of 238.16 words, demonstrating their rich vocabulary. For more detailed statistical data, please refer to Appendix A.1.

## 3 The Performance of LLMs on MenatQA

### 3.1 Task Definition

We focus on time-sensitive question answering tasks. The input of these tasks is formulated as $(c, q)$ for free-form generation tasks, where $c$ is the context and $q$ is the question. The desired output is either a span from the context or "unanswerable" text.

### 3.2 Baselines

In this section, we introduce the temporal reasoning models and currently popular large language models. These serve as the main evaluation backbone for MenatQA, enabling us to assess the performance of mainstream large language models on three types of temporal factors.

The baselines in our experiments include: Big-Bird (Zaheer et al., 2020) and FiD[2] (Izacard and Grave, 2020), ChatGLM (6B) (Zeng et al., 2022), BLOOM (7.1B) (Scao et al., 2022), GPT-J (6B) (Wang and Komatsuzaki, 2021), GPT-NEOX (20B) (Black et al., 2022), OPT (6.7B and 13B) (Zhang

---

[2]Especially, We use the versions of BigBird and FiD that have been fine-tuned on the Natural Questions (NQ; Kwiatkowski et al. 2019) dataset.

et al., 2022), LLAMA (7B and 13B) (Touvron et al., 2023), ChatGPT (gpt-3.5-turbo; Brown et al. 2020). The detailed information about models can be found in the Appendix A.3.6.

### 3.3 Results and Analysis

In this section, we identify weaknesses in LLMs with respect to three temporal factors by analyzing the differences among various models.

To validate the impact of various time-sensitive factors that were overlooked in previous works on the temporal reasoning ability of large language models, we test the performance of aforementioned LLMs under three time factors on MenatQA, as shown in Table 2. In order to further comprehensively analyze the susceptibility of large language models to temporal biases, we compare the performance of LLMs on extraction and reasoning questions in the scope factor of MenatQA, as shown in Table 3. Based on the results, we can find that:

Firstly, analyzing the results in Table 2, it can be observed that **LLMs display varying sensitivities towards different time factors**. Notably, the counterfactual and scope factors exert the most significant impact on LLMs, as shown by the highlighted sections with yellow background in the table. Additionally, not all LLMs outperform FiD on every type of factor. For instance, when evaluating the performance of GPT-3.5-turbo on the counterfactual factor, it fails to surpass FiD, with F1 and EM scores of 34.69 and 27.66, respectively. These scores are significantly lower than the corresponding results achieved by FiD (F1: 45.79, EM: 34.03). Besides, none of the other LLMs demonstrate superiority over FiD across all temporal factors, except for LLama-13B. In conclusion, LLMs still have limitations in effectively processing implicit temporal information, as indicated by their inadequate performance and sensitivity to different temporal factors. Therefore, more research is needed to enhance the temporal understanding and reasoning capabilities of LLMs.

Secondly, in extraction type questions, the majority of LLMs (i.e., ChatGLM-6B, Bloom-7B1, OPT-Series and GPT-Series) cannot achieve satisfactory outcomes when compared with temporal reasoning models (i.e., BigBird and Fid), as shown in Table 3. The weakness of LLMs in temporal reasoning is more prominent in reasoning type questions, where all LLMs exhibit varying degrees of performance decline compared to extraction type questions. This

| Method | MenatQA w/scope | | MenatQA w/order | | MenatQA w/counterfactual | | MenatQA w/all factors | |
|---|---|---|---|---|---|---|---|---|
| | F1 | EM | F1 | EM | F1 | EM | F1 | EM |
| Temporal Reasoning Models | | | | | | | | |
| BigBird(NQ) | 35.31 | 23.72 | 41.84 | 29.62 | 40.84 | 28.22 | 35.81 | 24.22 |
| FiD(NQ) | 41.03 | 30.63 | 45.85 | 33.73 | 45.79 | 34.03 | 36.12 | 26.32 |
| Large Language Models | | | | | | | | |
| ChatGLM-6B | 21.49 | 6.11 | 23.99 | 6.01 | 19.20 | 3.61 | 13.77 | 3.81 |
| Bloom-7B1 | 25.69 | 13.33 | 32.34 | 16.94 | 29.34 | 12.62 | 24.32 | 9.62 |
| GPT-J-6B | 34.56 | 24.95 | 39.40 | 27.86 | 45.64 | 33.07 | **37.84** | 25.85 |
| GPT-Neox-20B | 36.23 | 25.55 | 41.61 | 27.86 | 45.73 | 30.56 | **37.81** | 24.55 |
| OPT-6.7B | 29.14 | 19.84 | 35.79 | 24.35 | 32.77 | 20.54 | 27.27 | 16.13 |
| OPT-13B | 37.30 | 26.85 | 45.17 | 32.06 | 42.32 | 25.95 | 35.93 | 21.94 |
| LLama-7B | **45.71** | **34.47** | **51.93** | **37.17** | **49.78** | **33.97** | **37.53** | 23.45 |
| LLama-13B | **52.13** | **41.58** | **64.45** | **52.40** | **53.56** | **41.48** | **39.80** | **28.66** |
| GPT-3.5-turbo | **47.34** | **37.78** | **51.20** | **38.38** | 34.69 | 27.66 | 27.99 | 24.45 |

Table 2: The performance of models on each time-sensitive factor in the MenatQA dataset. **Bold scores** indicate superior performance compared to FiD. The factor with the most significant impact (lowest performance) on individual model is highlighted with yellow as background color.

| Method | Scope factor in MenatQA | | | |
|---|---|---|---|---|
| | Extraction | | Reasoning | |
| | F1 | EM | F1 | EM |
| BigBird(NQ) | 43.38 | 30.13 | 35.31 | 23.72 |
| FiD(NQ) | 48.77 | 36.73 | 41.03 | 30.63 |
| LLMs | | | | |
| ChatGLM-6B | 24.97 | 6.21 | 21.49 | 6.11 |
| Bloom-7B1 | 31.90 | 17.33 | 25.69 | 13.33 |
| GPT-J-6B | 38.86 | 27.86 | 34.56 | 24.95 |
| GPT-Neox-20B | 43.71 | 30.76 | 36.23 | 25.55 |
| OPT-6.7B | 35.50 | 24.65 | 29.14 | 19.84 |
| OPT-13B | 45.10 | 32.46 | 37.30 | 26.85 |
| LLama-7B | **57.09** | **43.09** | **45.71** | **34.47** |
| LLama-13B | **65.55** | **53.41** | **52.13** | **41.58** |
| GPT-3.5-turbo | **52.52** | **39.08** | **47.34** | **37.78** |

Table 3: The performance of LLMs on extraction and reasoning questions in the scope factor of MenatQA. **Bold Scores** indicate a higher performance than FiD.

finding proves that **LLMs are highly susceptible to temporal biases, and their ability to reason about time relies on the specific temporal information provided in the question**.

Finally, **larger parameter sizes generally lead to a stronger temporal reasoning ability in the same series of LLMs**. (i.e., LLama-7B & LLama-13B; and OPT-6.7B & OPT-13B). This conclusion is consistent with previous works (Zhong et al., 2021; Wei et al., 2022) that LLMs with a larger number of parameters tend to exhibit better perfor-

mance.

## 4   Simple Investigations for Improvement

In order to handle the three types of time factors in MenatQA, this paper proposes scope prompting, counterfactual prompting and rerank prompting methods under the zero-shot settings. Since the scope prompting method is not universal (e.g., it causes the EM score of GPT-3.5-turbo to drop from 37.78 to 31.36, as shown in Table 4), this paper explores tool learning and designs a time comparison tool specifically to address the scope factor questions.

### 4.1   Specific Prompts for Three Temporal Factors

**Base Prompt**  To evaluate the temporal reasoning performance of LLMs in the zero-shot setting, this paper uses the Base Prompt:

> **Base prompt**
>
> **Instruction:**
> Get answers for the question based on the context, where answers derived from substrings in the context or categorized as "unanswerable":
>
> **Context:** $\{c\}$  **Question:** $\{q\}$  **Answer:**

**Scope Prompt**  Following the way humans answer time scope questions, we first identify the start and

end time specifiers of the events in the context, and then compare the time in the question with the time interval of the corresponding event, so as to achieve temporal reasoning by comparing two time scopes. The scope prompting template is as follows:

> **Scope prompt**
>
> **Instruction:**
> Get answers for the question based on the context. If the time interval of when the event mentioned in the question occurred in the context, the answer is the span in the context. Else output "unanswerable" :
>
> **Context:** $\{c\}$  **Question:** $\{q\}$  **Answer:**

**Counterfactual Prompt** In this paper, we propose to transform the context to a narrator's statement and the question to enquire about the narrator's opinion in this statement (Zhou et al., 2023). Our method is motivated by our own cognitive process for answering different types of questions. The counterfactual prompting template is as follows:

> **Counterfactual prompt**
>
> **Instruction:** $\{Instruction\}$
> Eric reads an article as follows: $\{c\}$ Eric imagines counterfactuals that $\{h\}$
>
> **Question:** $\{q\}$ in Eric's imagination ?
> **Answer:**

**Rerank Prompt** In real-world scenarios, numerical information such as years often appears in different sentences in the text. For example, the recording of events is usually in chronological order, and the time specifier is used to distinguish different events. Therefore, we use the year information in the sentences to reorder the chronological sequence of multiple events. The rerank prompting template is as follows:

> **Rerank prompt**
>
> **Instruction:**
> Sort the sentences in the context by year, output the sorted context :
>
> **Context:** $\{c\}$
>
> **Sorted Context:**

In all of the above prompting templates, where $c$ denotes the context, $h$ represents the hypothetical

scenarios, and $q$ represents the main question of the original question. Specially, the *instruction* setting in the counterfactual prompt is consistent with the base prompt.

## 4.2 Tool Learning for Temporal Scope Factor

Tools provide domain-specific knowledge and capabilities. By leveraging tools to address the weaknesses of LLMs in tasks that go beyond the realm of pure natural language, such as arithmetic calculation (Wei et al., 2023) and table-based question answering (Lei et al., 2022), we can effectively bridge the gap between language understanding and task-specific requirements, enabling LLMs to excel in a wider range of applications beyond traditional NLP tasks.

**Time Comparison Tool** This paper follows the REACT (Yao et al., 2022), which prompts an LLM to generate reasoning texts that break down complex problems into intermediate steps, and action texts that allocate NLP tools for solving these steps. One example is that a LLM can make a decision based on real-time problems to call a search engine and gather the latest internet information that is not present in the pre-training corpus, and return it to the user. Inspired by the efficacy of reasoning and acting with LLMs and NLP tools, we explore the integration of time comparison tool with LLMs. In our setting, we build our time comparison tool based on the langchain[3] framework. By comparing whether the event mentioned in the question falls within the temporal scope corresponding to the events in the context, this approach helps LLMs understand temporal scope knowledge, as shown in Figure 8.

## 5 Experimental Results

As shown in Table 4 and Table 5, our observations indicate that utilizing special prompting methods and a tool learning method for three temporal factors can enhance the performance of LLMs.[4]

**Effect of the Scope Prompt** We present results in Table 4, where the section with a yellow background represents the effect of the scope prompting method. The scope prompting method improves performance over LLama-7B and LLama-13B (+1.10 and +1.41 on EM metrics). However,

---

[3] https://github.com/hwchase17/langchain
[4] We select LLMs with billions (LLama-7B), tens of billions (LLama-13B), hundreds of billions (GPT-3.5-turbo) parameter scales as baselines for the proposed solutions in this paper.

| Method | MenatQA w/scope | | MenatQA w/order | | MenatQA w/counterfactual | | MenatQA w/all factors | |
|---|---|---|---|---|---|---|---|---|
| | F1 | EM | F1 | EM | F1 | EM | F1 | EM |
| LLama-7B | 45.71 | 34.47 | 51.93 | 37.17 | 49.78 | 33.97 | 37.53 | 23.45 |
| LLama-7B + Prompts | **46.59** | **35.57** | **53.57** | **38.34** | **56.21** | **42.48** | **46.34** | **34.07** |
| LLama-13B | 52.13 | 41.58 | 64.45 | 52.40 | 53.56 | 41.48 | 39.80 | 28.66 |
| LLama-13B + Prompts | 51.72 | **42.99** | **64.47** | 52.19 | **60.55** | **49.90** | **48.37** | **38.98** |
| GPT-3.5-turbo | 47.34 | 37.78 | 51.20 | 38.38 | 34.69 | 27.66 | 27.99 | 24.45 |
| GPT-3.5-turbo + Prompts | 42.41 | 31.36 | **51.66** | **38.61** | **42.69** | **33.87** | **36.72** | **30.76** |

Table 4: The effect of the various prompting methods, where the scope prompt, order prompt, and counterfactual prompt are represented by the background colors, Blue , Green and Red , respectively. Notably, the Orange background color is used to indicate the simultaneous use of the scope prompt, order prompt and counterfactual prompt.

| Method | MenatQA | | | |
|---|---|---|---|---|
| | Scope Factor | | All Factors | |
| | F1 | EM | F1 | EM |
| LLama-7B | 45.71 (0.00) | 34.47 (0.00) | 37.53 (0.00) | 23.45 (0.00) |
| LLama-7B + prompts | 46.59 (0.88) | 35.57 (1.10) | 46.34 (8.81) | 34.07 (10.62) |
| LLama-7B + tool + prompts | 46.90 (1.19) | 35.37 (0.90) | 44.67 (7.14) | 32.67 (9.22) |
| LLama-13B | 52.13 (0.00) | 41.58 (0.00) | 39.80 (0.00) | 28.66 (0.00) |
| LLama-13B + prompts | 51.72 (-0.41) | 42.99 (1.41) | 48.37 (8.57) | 39.98 (10.32) |
| LLama-13B + tool + prompts | 65.85 (13.72) | 55.03 (13.45) | 61.06 (21.26) | 51.80 (23.14) |
| GPT-3.5-turbo | 47.34 (0.00) | 37.78 (0.00) | 27.99 (0.00) | 24.45 (0.00) |
| GPT-3.5-turbo + prompts | 42.41 (-4.93) | 31.36 (-6.42) | 36.72 (8.73) | 30.76 (6.31) |
| GPT-3.5-turbo + tool + prompts | 47.71 (0.37) | 37.58 (-0.20) | 38.65 (10.66) | 32.87 (8.42) |

Table 5: The table shows a comparison between the time comparison tool and the scope prompt on the scope factor and all factors. In brackets, the differences from scores compared to the original LLMs.

it does not do as well on GPT-3.5-turbo, which significantly reduces the EM score (-6.42).

**Effect of the Counterfactual Prompt** Based on the results in Table 4, we can find that the counterfactual prompt exhibits the greatest improvement in LLMs compared to the other two methods, with an average increase of 7.71 in EM score. This indicates that transforming counterfactual events into the perspective of others can effectively assist LLMs in achieving counterfactual temporal associations and reasoning.

**Effect of the Rerank Prompt** Compared to the highlighted sections with yellow background in Table 4, it can be observed that the use of the rerank prompt exhibits only a minor improvement in the order factor, possibly due to the loss of information in the sorted context. We conduct an evaluation of the quality of the reordered context, and the results reveal that LLMs are not inclined to output

every word in the context verbatim but rather tend to reorganize their language output, as shown in A.4.

**Effect of the Time Comparsion Tool** One the one hand, the experimental results in Table 5 indicate that the time comparison tool has stronger robustness compared to the scope prompting method, with similar performance on LLama-7B, and the time comparison tool does not cause drastically performance degradation on GPT-3.5-turbo, unlike the scope prompting method. Besides, the time comparison tool significantly improved the performance on LLama-13B, these results demonstrate that the tool is more suitable for LLMs with larger parameters to address time scope questions compared to the scope prompting method. On the other hand, the performance difference between LLama-7B and LLama-13B shows that LLMs with larger parameter sizes have a stronger capacity for utiliz-

ing tools. However, the performance of GPT-3.5-turbo do not improve, possibly due to its incorrect understanding of the temporal feedback provided by the tool and the limited impact of the scope factor (e.g., EM metrics from 39.08 to 37.78), as shown in Table 3.

# 6 Related Work

There have been plenty of works to tackle the temporal reasoning task. Zhang and Choi (2021) was introduced to tackle open-domain time-sensitive question answering, with a particular emphasis on analyzing how answers differ based on extra-linguistic factors , such as the time of inquiry. Kasai et al. (2022) extended time question answering to scenarios where real-time news serves as context, and requested the model to retrieve the latest temporal evidence to answer the question. StreamingQA (Liska et al., 2022) introduced the first QA dataset and task for studying adaptation to new information over time in open and close-book settings with temporally non-overlapping training and evaluation sets. TimeQA (Chen et al., 2021) built the first dataset to investigate whether existing models can understand time-sensitive facts.

There are a few major differences between the aforementioned works and MenatQA : 1) MenatQA encompasses various temporal factors, such as the scope factor, order factor, and counterfactual factor, involving a significant amount of reasoning about implicit temporal information. This aspect of temporal reasoning ability, which is neglected by previous works, is the most important. 2) MenatQA is not only the first dataset designed specifically for evaluating the time understanding and reasoning capabilities of LLMs, but also provides some simple optimization methods and baseline comparisons, which offer valuable references for evaluating the time reasoning of LLMs in the future. 3) Considering the existence of hallucinations in generative models, we introduce unanswerable types to penalize the illusory outputs of LLMs in MenatQA. These unanswerable type questions are impossible for humans to answer as well, and enable a genuine assessment of whether LLMs truly grasp temporal knowledge.

One concurrent work (published on 15 Jun 2023) similar to ours is (Tan et al., 2023), which proposed a comprehensive probing dataset TEMPREASON to evaluate the temporal reasoning capability of language models. They also proposed a temporal span

extraction and time-sensitive reinforcement learning framework to improve the temporal reasoning capability of large language models. However, they only evaluated three models, T5-Large (780M), Flan-T5-Large (780M), and GPT-3.5-turbo (175B), and mainly focused on using fine-tuning to improve the time reasoning ability of T5-Large and Flan-T5-Large. Besides, the fine-tuning based improvement methods are not applicable to large language models, such as OPT-175B. Our work aims to evaluate the time reasoning capability of current mainstream LLMs on three time-sensitive factors, and conducts preliminary investigations to improve the current LLMs on different time factors by designing various specific prompts and tool learning.

# 7 Conclusion

In this paper, we propose a question answering dataset named Multiple Sensitive Factors Time QA (MenatQA). It is the first dataset containing multiple time-sensitive factors that can be used as an evaluation benchmark for assessing the time understanding and reasoning abilities of LLMs. We find that most LLMs fall behind smaller temporal reasoning models with different degree on three factors. Moreover, the parameter size of LLMs substantially influences their capacity for temporal reasoning. LLMs also demonstrate a significant vulnerability to temporal biases and depend heavily on the precise temporal information provided in questions when reasoning about time. Finally, we conduct some preliminary investigations into improving the current LLMs' performance on the three temporal factors by utilizing prompting method and tool learning method, which could be potential avenues for future research.[5]

## Limitations

The 2853 samples in MenatQA can only be used as a test set for evaluating LLMs, and the data size is not sufficient for fine-tuning the models. However, this limitation can be mitigated by utilizing previous temporal reasoning datasets. The improvement solutions proposed in this paper, including the time comparison tool, scope prompt, rerank prompt, and counterfactual prompt, cannot be used as a complete and mature framework for LLMs. Instead, they represent a preliminary investigation aimed

---

[5]The dataset and code are released in `https://github.com/weiyifan1023/MenatQA`

at improving the LLMs' performance in time reasoning. Due to hardware limitations, we do not evaluate LLMs that require loading weights with a scale of more than 20B in the tens of billions parameter range.

## Acknowledgements

This work was supported by the National Key R&D Program of China (2022ZD0160503) and the National Natural Science Foundation of China (No.62276264). This research was also supported by Meituan. We thank Kang Liu, Yuanzhe Zhang, Yifan Wei and Xiaoyan Yu for helpful discussions of the methods used in this paper and the conclusions we reach from our experiments. Additionally, we thank Jun Zhao, Yisong Su, Huanhuan Ma and Fangyu Lei for feedback on writing and presentation of results. This work was conducted while Xiaoyan Yu and Yisong Su were an intern student at Institute of automation, Chinese academy of science.

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

## A  Appendix

### A.1  Data statistics

In MenatQA, the order factor can be combined with other counterfactual and scope factors. Specifically, the scope factor type can be further classified into granularity operation, contraction operation, and expansion operation, as shown in section 2.1. We calculated the proportions of different question types under ordered and unordered contexts, as shown in Figure 3 and Figure 4. Additionally, we also calculated the proportions of answerable and unanswerable question types, and the results are shown in Figure 5 and Figure 6.

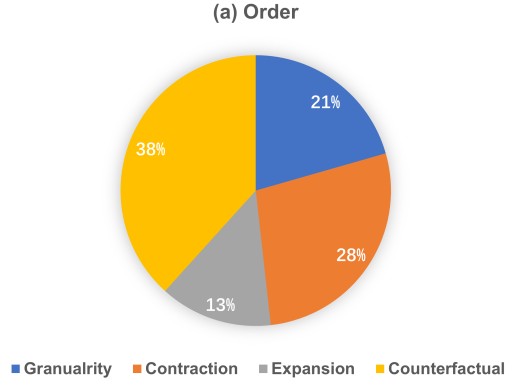

Figure 3: order statistic

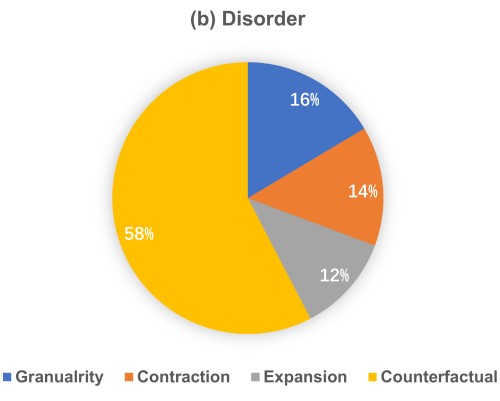

Figure 4: disorder statistic

### A.2  Data annotation

We recruit college students majoring in English-related fields and adopt the quality control approaches of annotator training and two-round validation to ensure the quality of MenatQA.

Considering the input length limitation of LLMs, we set the maximum number of documents in

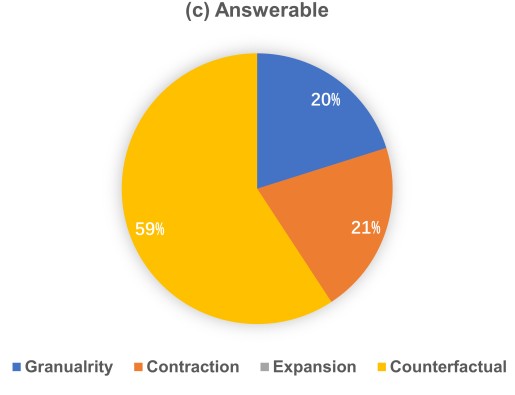

Figure 5: answerable statistic

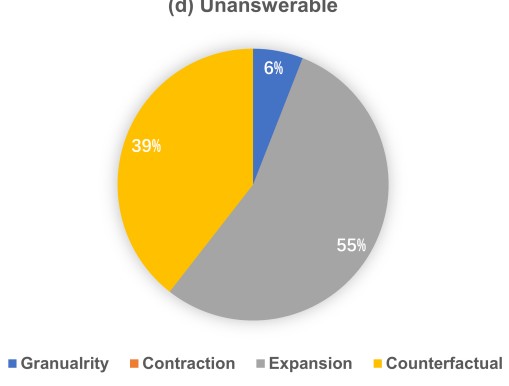

Figure 6: unanswerable statistic

TimeQA to 5, which already includes the gold evidence. Any other documents exceeding the maximum number will be filtered out. In our Closed Book QA setting, there is no need to set up a retriever to search for relevant documents. We ensure that all answers come from the context which provide based on the gold paragraph field given in TimeQA's annotation documents. Taking into account the phenomenon of knowledge conflicts , we restrict the temporal scope of the questions to before 2021. This measure ensures that the context from Wikipedia pages appears in the pretraining corpus of LLMs, thereby aligning the parameter knowledge of LLMs with the external context knowledge.

We generate *scope* factor type data using the prompt shown in Figure 7, *scope* factor can be further divided into three operations: *granularity*, *contraction*, and *expansion*.

## A.3    Experimental Setup

### A.3.1    Time Comparison Tool

The workflow diagram for using the time comparison tool is shown in Figure 8. In this paper, we used LLama-7B, LLama-13B, and GPT-3.5-turbo as the LLMs, and version 0.0.166 of the Langchain framework was used to implement the Time Comparison Tool.

### A.3.2    Zero-Shot Setting

All of our experiments were conducted under the zero-shot setting based on the base prompt, as shown in Base Prompt, and all the LLMs used in our experiments can be downloaded from the official website of Hugging Face.

### A.3.3    Extraction and Reasoning Questions

In section 3.3, to validate the sensitivity of LLMs to various time factors, only reasoning type questions were used in the scope factor, and extraction type questions were excluded. Sepcifically, extraction type questions originate from the TimeQA easy mode version, where the time points mentioned in the questions are explicitly present in the context. On the other hand, reasoning type questions involve time points that cannot be directly found in the context and require inference to obtain the answers. Moreover, reasoning type questions can be further classified into granularity questions, contraction questions, and expansion questions. These categories align with the classification in previous works, and therefore, no further discussion is needed.

### A.3.4    Baselines Setting

Based on the Table 2, we choose the best LLMs with parameter sizes at the billion scale (e.g., LLama-7B), tens of billions scale (e.g., LLama-13B), and hundreds of billions scale (e.g., GPT-3.5-turbo) as baseline models to evaluate the effectiveness of our proposed enhancement methods (e.g., Time Comparison Tool). To ensure that the predictions are consistent, we used the GPT-3.5-turbo-0301 version of ChatGPT.

### A.3.5    Parameter Setting

We use InstructGPT (gpt-3.5-turbo) as the frozen LLM, with temperature set to 0.0 and nucleus sampling set to 1 and n represents the number of chat completion options generated for each input prompt, which is set to 1. The hyperparameter settings for other LLMs are the same as above. We selected the EM metric as our primary evaluation metric to measure the performance of LLMs, and report performance averaged over 3 runs.

Narrow down the time scope in the question, and then update the question:
Question: Which employer did Philip Alston work for from 1996 to 2001?
Updated Question: Which employer did Philip Alston work for from 1997 to 2000?

Expand the time scope of the question, and then update the question:
Question: Which employer did Philip Alston work for from 1996 to 2001?
Updated Question: Which employer did Philip Alston work for from 1994 to 2002?

Add monthly information to the time scope of the question, and then update the question:
Question: Which employer did Philip Alston work for from 1996 to 2001?
Updated Question: Which employer did Philip Alston work for from August 1996 to July 2001?

Figure 7: few-shot construction prompt

### A.3.6 Baseline models

The models used in this paper are as follows:

- BigBird and FiD use 12 layers of encoder and decoder with 12 attention heads based on HugginFace Transformer.

- ChatGLM (6B), ChatGLM-6B is an open bilingual language model based on General Language Model (GLM) framework, with 6.2 billion parameters. ChatGLM-6B uses technology similar to ChatGPT, optimized for Chinese QA and dialogue.

- BLOOM (7.1B), BLOOM model is a large decoder-only language model pretrained for around 350 billion tokens with an architecture similar to GPT-3.

- GPT-J (6B), an auto-regressive text generation model trained on the Pile with 6 billion parameters.

- GPT-NEOX (20B), a 20 billion parameter auto-regressive language model trained on the Pile.

- OPT (6.7B and 13B), a suite of decoder-only pre-trained transformers ranging from 125M to 175B parameters.

- LLAMA (7B and 13B) , a collection of foundation language models ranging from 7B to 65B parameters, and it is possible to train

state-of-the-art models using publicly available datasets exclusively, without resorting to proprietary and inaccessible datasets. In particular, LLAMA-13B outperforms GPT-3 (175B) on most benchmarks.

- ChatGPT (gpt-3.5-turbo), the most capable and cost effective model in the GPT-3.5 family is gpt-3.5-turbo which is optimized for chat but works well for traditional completions tasks as well, and openai recommends using gpt-3.5-turbo over the other GPT-3.5 models because of its lower cost.

### A.4 Case Study

Compared to other LLMs, GPT-3.5-turbo tends to classify questions that require temporal reasoning (where the time period is not directly mentioned in the context) as unanswerable, leading to incorrect outputs of "unanswerable". On the other hand, other LLMs such as LLama are lacking in their ability to reject answering, making them more likely to classify the questions as answerable. Therefore, the impact of the counterfactual type on LLama is smaller compared to GPT-3.5-turbo.

Here are the possible reasons we have inferred: 1) LLMs exhibit performance degradation due to scope factors, as the time involved in the questions does not appear directly in the context and requires model inference to derive the answer. This type is more challenging compared to extractive questions, as shown in Table 3. 2) To the best of our knowledge, we are the first to introduce counterfactual

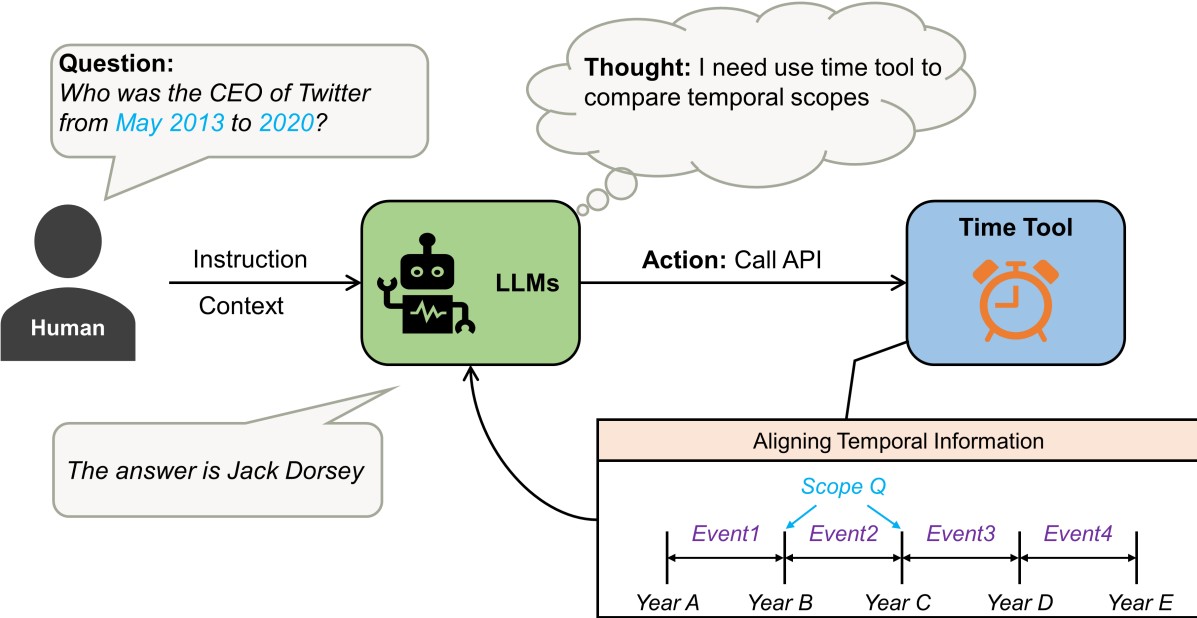

Figure 8: The overall process of the Time Comparison Tool. The Time Comparison Tool is used to determine whether the events in the question belong to the corresponding time range of the events in the context. Specially, *Scope Q* refers to the blue temporal information involved in the question, and the timeline represents the events that appear in the context and their corresponding time ranges.

questions related to time, which LLMs have not encountered in their pretraining corpus, resulting in poor performance on these counterfactual questions. 3) Furthermore, in the order type, since the time relevant to the question appears directly in the context, LLMs can directly extract the answer from the context, resulting in the least performance degradation on this factor.

## A.5 Sample Introduction

In this work, unanswerable questions refer to questions that cannot be inferred from the temporal information in the context within a given time period. As shown in Figure 1, an example of unanswerable question is "who was the CEO of Twitter from 2005 to March 2006?". In cases like this, where the time period mentioned in the question is not addressed in the context, it is not possible to infer and provide an answer. Therefore, the appropriate output would be marked as "unanswerable." The sample results of the rerank prompt are shown in Figure 9.

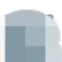

Instruction: Sort the sentences in the context by year, output the sorted context:
Context: Attaphol began his career as a player at Thai Port FC Authority of Thailand in 1985 . His time from 1991 to 1994 was marked by less success than in his first stay at Port Authority. From 1994 to 1996 he played for Pahang again and this time. In his first year , he won his first championship with the club . He played for the club until 1989 and in 1987 also won the Queens Cup . He then moved to Malaysia for two seasons for Pahang FA , then return to Thailand to his former club .

Sorted Context:

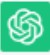

Sorted Context:

Attaphol began his career as a player at Thai Port FC Authority of Thailand in 1985.

He played for the club until 1989 and in 1987 also won the Queens Cup.

From 1991 to 1994 was marked by less success than in his first stay at Port Authority.

He then moved to Malaysia for two seasons for Pahang FA, then returned to Thailand to his former club.

From 1994 to 1996, he played for Pahang again and this time, in his first year, he won his first championship with the club.

Figure 9: Rerank case on MenatQA using rerank prompt.