# OpenReview forum: "MenatQA: A New Dataset for Testing the Temporal Comprehension and Reasoning Abilities of Large Language Models"
_EMNLP/2023/Conference — EMNLP 2023 Findings_

### Official Review · Reviewer_7Azk · 2023-07-24

**Soundness:** 4

**Excitement:**

3: Ambivalent: It has merits (e.g., it reports state-of-the-art results, the idea is nice), but there are key weaknesses (e.g., it describes incremental work), and it can significantly benefit from another round of revision. However, I won't object to accepting it if my co-reviewers champion it.

**Paper Topic And Main Contributions:**

Temporal reasoning ability is an important ability of LLMs. Existing dataset mainly considers the temporal scope factor but neglecting other temporal reasoning abilities (factors). This paper propose a new dataset named MenatQA which is to evaluate the time comprehension and reasoning abilities of LLMs. This dataset contains three temporal factors (scope factor, order factor, counterfactual factor) with total 2,853 samples. Based on the proposed MenatQA, the conducted experiments on several mainstream models by using specific prompts and tool learning. The results demonstrate that the majority of LLMs perform poorly on the MenatQA dataset.

**Reasons To Accept:**

This resource paper propose a useful dataset to evaluate the temporal reasoning ability of LLMs. They conducted preliminary investigations to optimise temporal reasoning ability of LLMs, which can be used as baseline to inspire the future research. I suggest this paper should be accepted into Findings.

**Reasons To Reject:**

Not much technical innovation. This resource paper is more suitable to be accepted into Findings.

**Reproducibility:**

4: Could mostly reproduce the results, but there may be some variation because of sample variance or minor variations in their interpretation of the protocol or method.

**Reviewer Confidence:**

3: Pretty sure, but there's a chance I missed something. Although I have a good feel for this area in general, I did not carefully check the paper's details, e.g., the math, experimental design, or novelty.

---

> ### Author Rebuttal · Authors · 2023-08-26
>
> **Weakness:**  Not much technical innovation. This resource paper is more suitable to be accepted into Findings.
>
>
> - In this paper, we found that existing LLMs lag behind in temporal reasoning ability. We propose the MenatQA dataset as a benchmark specifically desingned to **evaluate LLMs' temporal reasoning ability**. To the best of our knowledge, **we are the first to introduce** multiple factors for temporal reasoning of LLMs. On the **technical front**, we propose simple yet effective method that can **serve as future baselines** for evaluating temporal reasoning capabilities of LLMs.
>
> - Our technical innovation is two-fold. First, we introduce a time comparison tool and employ tool learning techniques specifically tailored for temporal reasoning tasks. The design of tools for LLMs and the instruction of LLMs on their usage represent a hot research area. Our work **not only fills the gap in tool learning for such tasks** but also **provides LLMs with a practical temporal reasoning tool that can be readily employed**. Second,  we designed prompt methods specifically tailored for temporal reasoning, targeting three different time factors. This is **the first prompt-based method for temporal reasoning tasks**.

---

### Official Review · Reviewer_2UnL · 2023-08-04

**Soundness:** 2

**Excitement:**

2: Mediocre: This paper makes marginal contributions (vs non-contemporaneous work), so I would rather not see it in the conference.

**Paper Topic And Main Contributions:**

This paper aims to assess the performance of current Language Models (LLMs) on Temporal Comprehension and Reasoning.

**Questions For The Authors:**

1. TimeQA is developed for time-sensitive question answering, specifically targeting the temporal scope factor. On the other hand, this paper introduces MenatQA, which also emphasizes the scope factor. So, what are the distinctions between the two datasets?

2. The authors mention that they utilize a two-round validation process to ensure the quality of MenatQA. However, the specific details regarding this two-round validation are not adequately explained.

3. It would be beneficial to provide some examples of unanswerable questions.


**Reasons To Accept:**

Evaluating the abilities of LLMs on Temporal Comprehension and Reasoning is a meaningful task.


**Reasons To Reject:**

1. This paper aims to evaluate the performance of current LLMs on different temporal factors and select three types of factors, including cope, order, and counterfactual. What is the rationale behind selecting these three types of factors, and how do they relate to each other?

2. More emphasis should be placed on prompt design. This paper introduces several prompting methods to address issues in MenatQA. Since different prompts may result in varying performance outcomes, it is essential to discuss how to design prompts effectively.

3. The analysis of experimental results is insufficient. For instance, the authors only mention that the scope prompting method shows poor performance on GPT-3.5-turbo, but they do not provide any analysis of the underlying reasons behind this outcome.


**Reproducibility:**

2: Would be hard pressed to reproduce the results. The contribution depends on data that are simply not available outside the author's institution or consortium; not enough details are provided.

**Reviewer Confidence:**

4: Quite sure. I tried to check the important points carefully. It's unlikely, though conceivable, that I missed something that should affect my ratings.

---

> ### Author Rebuttal · Authors · 2023-08-26
>
> **Weakness 1**: What is the rationale behind selecting these three types of factors, and how do they relate to each other?
>
> - **The rationale behind selecting these three types of factors is detailed in the 3rd paragraph in Section 1.** For your better understanding, here is the rationale behind each factors:
>   * Scope factor: time-related questions require inference of time that is not explicitly stated in the context, infering such scope is important for temporal reasoning.
>   * Order factor: the description of events typically follows a chronological order, recognizing such order is a fundamental ability concerning temporal reasoning.
>   * Counterfactual factor: questions with temporal assumptions greatly escalate the difficulty of temporal reasoning, answering such questions require counterfactual thinking of models.
>
> - They are all important factors in temporal reasoning, each fulfilling distinct roles and serving specific purposes. While **they may not necessarily be interrelated**, they collectively contribute to the overall process of temporal reasoning.
>
> **Weakness 2**: Since different prompts may result in varying performance outcomes, it is essential to discuss how to design prompts effectively.
>
> - The main contribution of the paper is to propose a new dataset to verify the temporal understanding and reasoning abilities of LLMs on three factors. We just proposed three simple prompts to improve such abilities of LLMs on the proposed dataset，which are very simple and could be regarded as the baselines for future researches. But we agree with you that designing effective prompts is very important for improving performance. We will leave it as the next research problem. Thanks for your suggestions.
>
> **Weakness 3**: The analysis of experimental results regarding the poor performance of GPT-3.5-turbo is insufficient.
>
> Thanks for your valuable suggestion. We agree that the reason of the poor performance of GPT-3.5-turbo should be explored. We believe this issue is the same as mentioned by Reviewer LWDP, and **please see response to Reviewer 1**.
>
> - Regarding the reasons behind the poor temporal reasoning capabilities of LLMs, we conducted relevant experiments for analysis. For example, in the case study, we randomly sampled 100 outputs from GPT-3.5-turbo. We found that the use of scope prompts led to subpar performance of GPT-3.5-turbo. This was because **the scope prompt made GPT-3.5-turbo more inclined to consider scope type questions as unanswerable**, resulting in incorrect outputs labeled as "unanswerable."
>
> We apologize for the lack of detailed discussions in the experiments. We will provide supplements to the case study section in subsequent revisions to address this issue. Furthermore, **it is unfeasible to delve deeper into the underlying reasons due to the fact that GPT-3.5-turbo has not been open-sourced**.
>
>
> **Question 1**:  what are the distinctions between TimeQA and MenatQA ?
>
> - While both TimeQA and MenatQA are temporal reasoning datasets, they have distinct objectives. MenatQA is designed to assess the temporal reasoning ability of LLMs.
> - Compared with MenatQA, TimeQA only addressed the scope type. Specifically MenatQA focused not only on the scope type, but also on the counterfactual and order types. Even though on the scope type, MenatQA further provides more detailed categorization such as granularity, contraction, and expansion. By contrast, in TimeQA, these distinctions are not explicitly made, except for the mention of granularity (month) as a specific case.
>
>
> **Question 2**: The specific details regarding this two-round validation are not adequately explained.
>
> - The two-round validation process, as **outlined in Appendix A.2**, follows the same procedure to other works involving data annotation. To **provide a clearer understanding**, here is a general overview of the two-round validation process:
>      * Round 1: Annotators are trained on the annotation guidelines and provided with sample data, and then independently annotate a subset of the dataset based on the guidelines. A quality inspector checks for consistency, adherence to guidelines, and potential errors or ambiguities. Any discrepancies or issues are identified and discussed with the annotators.
>      * Round 2: Annotators revise and refine their annotations based on the feedback and clarifications received. The quality inspector compares the Round 1 and Round 2 annotations, looking for consistency and improvements. In case of discrepancies or disagreements, further discussions and consensus-building take place.
>
> **Question 3**: It would be beneficial to provide some examples of unanswerable questions.
>
> + In this work, unanswerable questions refer to questions that cannot be inferred from the temporal information in the context within a given time period.
> + As shown in Figure 1, an example of unanswerable question is "who was the CEO of Twitter from 2005 to March 2006?". In cases like this, where **the time period mentioned in the question is not addressed in the context, it is not possible to infer and provide an answer**. Therefore, the appropriate output would be marked as "unanswerable."
> + We will add examples of unanswerable questions to the appendix in the revision.

---

### Official Review · Reviewer_LWDP · 2023-08-05

**Soundness:** 3

**Excitement:**

3: Ambivalent: It has merits (e.g., it reports state-of-the-art results, the idea is nice), but there are key weaknesses (e.g., it describes incremental work), and it can significantly benefit from another round of revision. However, I won't object to accepting it if my co-reviewers champion it.

**Paper Topic And Main Contributions:**

This paper presents a new dataset for testing the time comprehension and reasoning abilities of large language models (LLMs), they consider three temporal factors, i.e., scope factor, order factor, and counterfactual factor. After testing multiple models on this dataset, they find that LLMs may fall behind smaller temporal reasoning models. Also, scope factor and counterfactual factor generally impact more on LLMs, and LLMs struggle more with reasoning compared with extractions. This paper also proposes different prompting methods and a tool learning method to improve the model performance.

**Questions For The Authors:**

A. Time understanding and reasoning is related to the common sense of month order and math computations. I'm curious whether adding instructions regarding month order helps, and whether Chain of Thought prompting helps the model.

**Reasons To Accept:**

The proposed dataset might be useful to inspire further studies regarding time comprehension and reasoning abilities. The experiments and analyses are comprehensive. The proposed prompt designs and including time comparison tools work well.

**Reasons To Reject:**

Although the creation and analysis of the dataset are commendable, it would be beneficial if the authors could further analyze when and why the model makes mistakes, for now, it doesn't significantly advance our understanding beyond existing knowledge.

**Reproducibility:**

4: Could mostly reproduce the results, but there may be some variation because of sample variance or minor variations in their interpretation of the protocol or method.

**Reviewer Confidence:**

3: Pretty sure, but there's a chance I missed something. Although I have a good feel for this area in general, I did not carefully check the paper's details, e.g., the math, experimental design, or novelty.

---

> ### Author Rebuttal · Authors · 2023-08-26
>
> **Weakness 1**: Although the creation and analysis of the dataset are commendable, it would be beneficial if the authors could further analyze when and why the model makes mistakes, for now, it doesn't significantly advance our understanding beyond existing knowledge.
>
> - Thanks for your valuable suggestion. We agree that the analysis of "when and why the model makes mistakes" will enhance the overall comprehensiveness of our research. Regrettably, **it is unfeasible to provide additional analysis on the fundamental reasons behind LLMs, as it involves factors such as the structure of different LLMs, pretraining methods, and pretraining corpora**. However, we will provide preliminary results through the case study for further analysis in the revision.
>
>
>  - Compared to other LLMs, GPT-3.5-turbo **tends to classify questions that require temporal reasoning** (where the time period is not directly mentioned in the context) **as unanswerable**, leading to incorrect outputs of "unanswerable". On the other hand, other LLMs such as LLama are **lacking in their ability to reject answering**, making them more likely to classify the questions as answerable. Therefore, the impact of the counterfactual type on LLama is smaller compared to GPT-3.5-turbo.
>
> - Here are the possible reasons we have inferred:
>     1.  LLMs exhibit performance degradation due to scope factors, **as the time involved in the questions does not appear directly in the context** and **requires model inference to derive the answer**. This type is more challenging compared to extractive questions, as shown in Table 3.
>     2. To the best of our knowledge, we are the first to introduce **counterfactual questions related to time**, which **LLMs have not encountered in their pretraining corpus**, resulting in poor performance on these counterfactual questions.
>     3. Furthermore, in the order type, **since the time relevant to the question appears directly in the context**, LLMs can directly extract the answer from the context, resulting in the least performance degradation on this factor.
>
> To validate the aforementioned reasoning, we can conduct additional experiments by fine-tuning the FID model  on three time-sensitive factors. This will help verify if the observed results are due to pretraining bias. To procure a more comprehensive understanding and gain advanced insights,  we will continue to investigate this matter in subsequent research endeavors.
>
> **Question A**: whether adding instructions regarding month order helps, and whether Chain of Thought prompting helps the model?
>
> 1. **We believe that adding instructions regarding month order would not be effective.**  In the case of GPT-3.5-turbo, simple experiements show that it possesses commonsense knowledge regarding the order of months in a year. Consequently, we believe that adding instructions specifically addressing the month order would not yield effective results. Furthermore, in the experiments of Time Comparison Tool (Section 4.2), we used a Python script to compare the time information (Year-Month) extracted by LLMs, **ensuring the correctness of time comparison**. However, the results indicate no improvement (Table 5), which suggests that adding instructions regarding the month order provide no benefit.
>
>
>
> 2. **We believe that adding CoT-based prompts will be helpful for improving temporal reasoning of LLMs**. However, considering **the limitations of context length and LLMs' input length**, our experiments were conducted **in a fair zero-shot setting**. This idea is worthy of further exploration.

---

### Meta-Review · Area_Chair_c3Wo · 2023-09-19

**Recommendation:** 3

**Metareview:**

This paper presents a new dataset, MenatQA, for testing the time comprehension and reasoning abilities of large language models (LLMs). The reviewers appreciate the comprehensive experiments and analyses, and the proposed prompt designs. However, they suggest that the authors could further analyze when and why the model makes mistakes, the rationale behind a few design choices, limitations in technical innovations. The authors' rebuttal highlights the value of the work, provides some explanation for the the experimental design, and explains some limitations, but there are still some opaqueness in the analysis and also limited novelty.

---

### Decision · Program_Chairs · 2023-10-07

**Decision:**

Accept-Findings

**Comment:**

This paper presents a new dataset, MenatQA, for testing the time comprehension and reasoning abilities of large language models (LLMs). The reviewers appreciate the comprehensive experiments and analyses, and the proposed prompt designs. However, they suggest that the authors could further analyze when and why the model makes mistakes, the rationale behind a few design choices, limitations in technical innovations. The authors' rebuttal highlights the value of the work, provides some explanation for the the experimental design, and explains some limitations, but there are still some opaqueness in the analysis and also limited novelty.